# 3DRealCar: An In-the-wild RGB-D Car Dataset with 360-degree Views

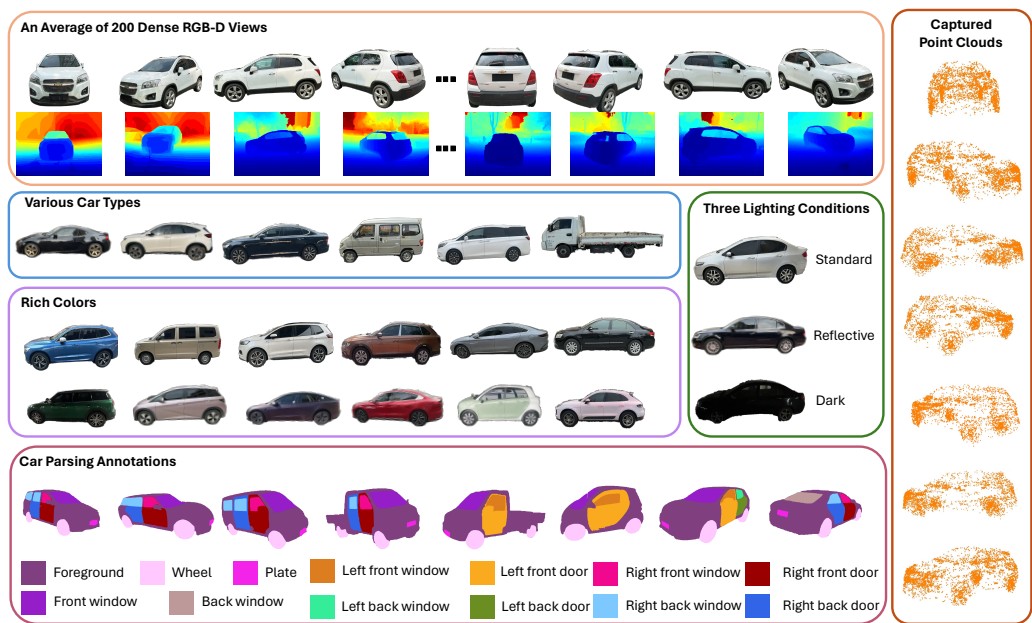

Figure 1: **Characteristics of our curated high-quality 3DRealCar dataset.** 3DRealCar contains detailed annotations for various colors, car types, brands, and even car parsing maps. In particular, our dataset contains three lighting conditions on car surfaces, bringing challenges to existing methods.

## Abstract

3D cars are commonly used in self-driving systems, virtual/augmented reality, and games. However, existing 3D car datasets are either synthetic or low-quality, presenting a significant gap toward the high-quality real-world 3D car datasets and limiting their applications in practical scenarios. In this paper, we propose the first large-scale 3D real car dataset, termed 3DRealCar, offering three distinctive features. (1) **High-Volume**: 2,500 cars are meticulously scanned by smartphones, obtaining car images and point clouds with real-world dimensions; (2) **High-Quality**: Each car is captured in an average of 200 dense, high-resolution 360-degree RGB-D views, enabling high-fidelity 3D reconstruction; (3) **High-Diversity**: The dataset contains various cars from over 100 brands, collected under three distinct lighting conditions, including reflective, standard, and dark. Additionally, we offer detailed car parsing maps for each instance to promote research in car parsing tasks. Moreover, we remove background point clouds and standardize the car orientation to a unified axis for the reconstruction only on cars and controllable rendering without background. We benchmark 3D reconstruction results with state-of-the-art methods across each lighting condition in 3DRealCar. Extensive experiments demonstrate that the standard lighting condition part of 3DRealCar can be used to produce a large number of high-quality 3D cars, improving various 2D and 3D tasks related to cars. Notably, our dataset brings insight into the fact that recent 3D reconstruction methods face challenges in reconstructing high-quality 3D cars under reflective and dark lighting conditions. Our dataset is available here.

Table 1: **The comparison of existing 3D car datasets.** Our dataset contains unique characteristics compared with existing 3D car datasets. Lighting means the lighting conditions of the surfaces of cars. Point Cloud represents the point clouds with actual sizes in real-world scenes.

| Dataset | Instances | Type | Views | Resolution | Brand | Lighting | Car Parsing | Depth | Point Cloud |
|---|---|---|---|---|---|---|---|---|---|
| SRN-Car | 2151 | Synthetic | 250 | 128×128 | ✗ | ✗ | ✗ | ✗ | ✗ |
| Objaverse-car | 511 | Synthetic | - | - | ✗ | ✗ | ✗ | ✗ | ✗ |
| MVMC | 576 | Real | ∼10 | 600×450 | ∼40 | ✗ | ✗ | ✗ | ✗ |
| **3DRealCar (Ours)** | **2500** | **Real** | **∼200** | **1920×1440** | **100+** | **3** | **13** | ✓ | ✓ |

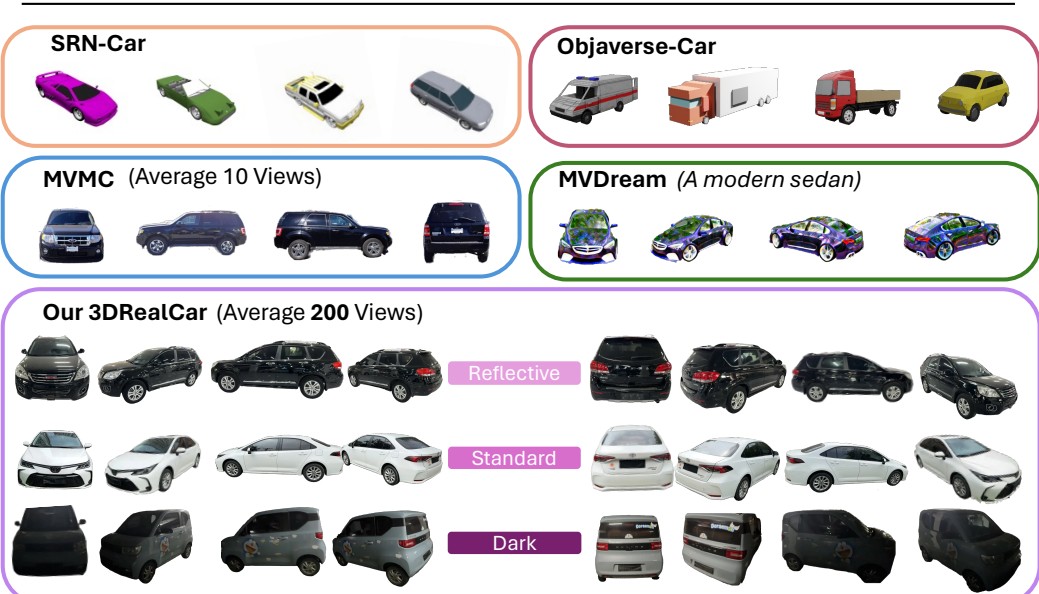

Figure 2: **Visual comparisons of 3D car datasets and the results of a 3D generative method.** Our 3DRealCar is captured in real-world scenes and contains more densely captured views. In addition, our dataset has annotations for three different lighting conditions on the car surface. We also compare a recent state-of-the-art text-to-3D model, MVDream (Shi et al., 2023b) with a prompt "*a modern sedan*", demonstrating its failure to generate high-quality 3D car models.

## 1 INTRODUCTION

Cars, as both daily objects and vehicles, are of significant interest to researchers, especially in the field of autonomous driving. Autonomous perception systems are typically trained on daily scene datasets that are collected frequently. However, these datasets often exhibit long-tailed distributions, with far fewer instances of corner-case scenarios, like car accidents. Consequently, this imbalance leads to the autonomous perception system generalizing well in the most frequently occurring scenes. This means that the system is likely to perform poorly in rare situations, posing significant safety risks to drivers. To build a reliable system, it is essential to have a simulator that can simulate photorealistic hazardous scenes. Moreover, high-quality 3D cars are necessary for a realistic simulator.

Recent 3D car reconstruction methods (Wang et al., 2023a; Zhou et al., 2023; Xie et al., 2023) mainly reconstruct cars from self-driving datasets (Sun et al., 2020; Caesar et al., 2020; Geiger et al., 2013). To apply reconstructed cars to real-world scenes, the reconstructed 3D car should be high-quality. However, it is very challenging to obtain such high-quality 3D cars for the following reasons: (1) Previous 3D car reconstruction methods produce low-quality 3D cars, primarily because they train on self-driving datasets with low-resolution car images and a limited number of trainable views. (2) Manually crafting a high-quality 3D car model requires specialized artists, which is time-consuming. (3) There is no large-scale 3D real car dataset that can be utilized to produce a bulk of 3D cars.

Moreover, existing 3D car datasets are either synthetic or only contain a few posed images, as shown in Figure 2. SRN-Car (Chang et al., 2015) and Objaverse-Car (Deitke et al., 2023) collect 3D car

computer-aided design (CAD) models from the Internet, but these models are synthetic and contain non-photorealistic texture. Although MVMC (Zhang et al., 2021) is a real car dataset, it collects only ten views on average for each car. On the contrary, our collected 3DRealCar dataset provides an average of 200 dense RGB-D views per car for high-quality 3D car reconstruction.

We also show that the recent state-of-the-art 3D generative method, MVDream (Shi et al., 2023b), as depicted in Figure 2, fails to generate high-quality cars due to the multi-view inconsistency introduced by generative models (Rombach et al., 2022; Stability.AI, 2023; Liu et al., 2023c; Sun et al., 2023). Thus, the existing 3D generation methods cannot be employed to generate high-quality 3D real car assets.

In this work, we collect a large-scale 3D real car dataset in the wild, termed 3DRealCar, which contains dense high-quality views and rich diversity. During data collection, we employ smartphones with ARKit (Apple, 2021) to scan cars parked on roadsides or parking lots, obtaining posed RGB-D images and point clouds of cars. In particular, we scan around the cars in three loops to obtain dense views. Note that we collect car data with the consent of owners. In Table 1 and Figure 1, we show our dataset possesses striking characteristics compared with previous 3D car datasets. We capture dense RGB-D images in high resolution, which promotes the reconstruction of high-quality 3D cars. Furthermore, we scan cars under three different lighting conditions, resulting in the surfaces of cars having different lighting effects, such as reflective, standard, and dark, where we denote the standard as the smooth lighting condition without obvious specular highlight. Figure 2 shows some examples of three lighting conditions in our dataset. Note that the number of instances in our dataset is the largest in existing datasets. Therefore, our collected 3DRealCar dataset has a rich diversity in terms of car types, colors, brands, and lighting conditions. We also provide car parsing map annotations with thirteen classes for each instance, which enable our dataset to be applied in car component understanding tasks.

To construct a high-quality dataset, we filter out the images that are out of focus, occluded, or blurred. To facilitate the 3D reconstruction solely on cars, we remove the point clouds of the background. We also adjust the orientation of the car facing along the x-axis before the reconstruction for controllable rendering. Based on the high-quality posed RGB-D images, point clouds, and multi-grained annotations, we can apply the dataset to various tasks related to cars. Figure 3 shows our dataset supports over 10 tasks, including several popular 2D and 3D tasks to promote the advancement of car-related research.

We leverage existing state-of-the-art methods to benchmark 3D reconstruction and car parsing tasks of our 3DRealCar dataset. We also conduct extensive experiments to demonstrate that the reflective and dark lighting conditions in our dataset are challenging to existing methods, which brings a new challenge for 3D reconstruction in awful lighting conditions. Furthermore, we demonstrate that our 3DealCar dataset can bring real-car prior and enhance existing 3D generation and downstream methods. Overall, the contributions of this work can be summarized below:

• We propose the first large-scale 3D real car dataset, named 3DRealCar, which contains 2,500 car instances and their point clouds with actual sizes in real-world scenes.

• 3DRealCar contains RGB-D images and point clouds with detailed annotations, supporting researchers to investigate various tasks in both 2D and 3D.

• We conduct 3D reconstruction and car parsing benchmarks to advance car-related tasks. Notably, we observe that existing methods face challenges under the extreme lighting conditions of 3DRealCar.

• Extensive experiments demonstrate our 3DRealCar dataset can enhance real-car prior and improve the performance of existing 3D generation and novel view synthesis methods.

## 2  RELATED WORK

**3D Car Datasets.** There are several well-known large-scale autonomous driving datasets so far, such as Nuscenes (Caesar et al., 2020), KITTI (Geiger et al., 2013), Waymo (Sun et al., 2020), Pandaset (Xiao et al., 2021), ApolloScape (Huang et al., 2018), and Cityscape (Cordts et al., 2016). These datasets are captured by multi-view cameras and lidars mounted on ego cars. Various works (Wang et al., 2023a; García Orellana et al., 2001; Liu et al., 2024; Xie et al., 2023) attempt to reconstruct 3D cars in these datasets. However, these methods fall short of reconstructing high-

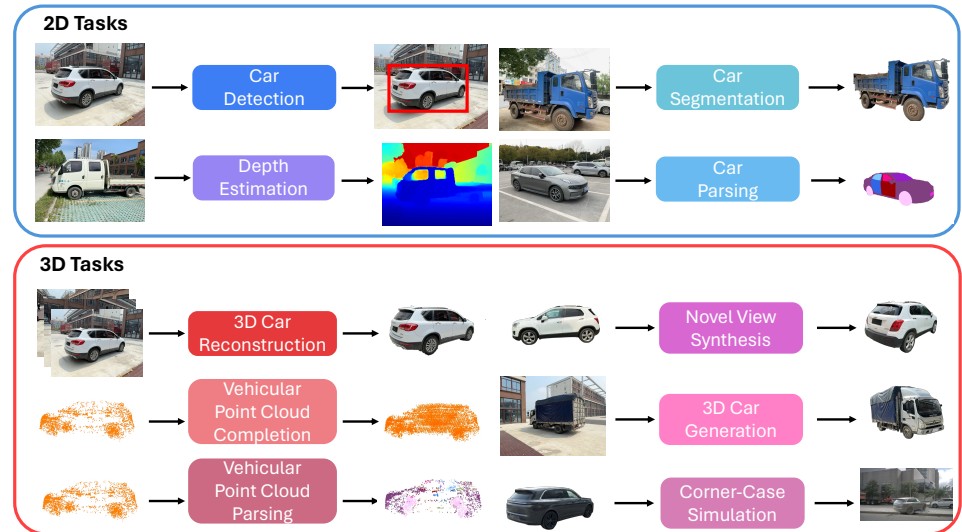

Figure 3: **The applicable tasks of our dataset.** Our proposed 3DRealCar dataset containing RGB-D images, point clouds, and rich annotations can be applied to various popular 2D and 3D tasks to support the construction of safe and reliable self-drving system.

quality 3D cars due to the lack of sufficient and dense training views. DeepMANTA (Chabot et al., 2017) provides car component segmentation maps, but this dataset is based on the synthetic CAD model that cannot be accurately used in real-world settings. SRN-Car (Chang et al., 2015) and Objaverse (Deitke et al., 2023) collect 3D car models from existing repositories and Internet sources. However, these datasets only contain synthetic cars, which cannot produce realistic textures and geometry. MVMC (Zhang et al., 2021) is collected from car advertising websites, which contain a series of car images, especially multi-view images of each car. However, the views of images per car in MVMC are unposed and sparse, which is adverse to reconstructing high-quality 3D car models. In this paper, we collect a high-quality 3D real car dataset to fill the above gaps.

**3D Reconstruction with Neural Field.** 3D reconstruction aims to create a 3D structure digital representation of an object or a scene from its multi-view images, which is a long-standing task in computer vision. One of the most representative works in 3D reconstruction is Neural Radiance Fields (NeRFs) (Mildenhall et al., 2021), which demonstrates promising performance for novel view synthesis. Afterward, this method inspires a new wave of 3D reconstruction methods using the volume rendering method, with subsequent works focusing on improving its quality (Verbin et al., 2021; Barron et al., 2021; 2022b; Guo et al., 2022; Suhail et al., 2022; Chen et al., 2022b; Wang et al., 2023b; Barron et al., 2023), efficiency (Fridovich-Keil et al., 2022; Müller et al., 2022; Reiser et al., 2021; Sun et al., 2022; Kerbl et al., 2023a; Chen et al., 2022a; Garbin et al., 2021), applying artistic effects (Fan et al., 2022; Wang et al., 2022; Jain et al., 2022; Zhang et al., 2022b), and generalizing to unseen scenes (Yu et al., 2021; Chen et al., 2021; Wang et al., 2021; Johari et al., 2022; T et al., 2023; Chibane et al., 2021). Particularly, Kilonerf (Reiser et al., 2021) accelerates the training process of NeRF by dividing a large MLP into thousands of tiny MLPs. Furthermore, Mip-NeRF (Barron et al., 2021) proposes a conical frustum rather than a single ray to ameliorate aliasing. Mip-NeRF 360 (Barron et al., 2022a) further improves the application scenes of NeRF to the unbounded scenes. Although these NeRF-based methods demonstrate powerful performance on various datasets, the training time always requires several hours even one day more. Instant-NGP (Müller et al., 2022) uses a multi-resolution hash encoding method, which reduces the training time by a large margin. 3DGS (Kerbl et al., 2023a) proposes a new representation based on 3D Gaussian Splatting, which reaches real-time rendering for objects or unbounded scenes. 2DGS (Huang et al., 2024) proposes a perspective-accurate 2D splatting process that leverages ray-splat intersection and rasterization to further enhance the quality of the reconstructions. Scaffold-GS (Lu et al., 2023) proposes an anchor growing and pruning strategy to accelerate the scene coverage, which effectively reduces redundant Gaussians and improves rendering quality. However, there is not yet a large-scale 3D real car dataset so far. Therefore, we present a 3D real car dataset, named 3DRealCar in this work.

Figure 4: **Illustration of our data collection and preprocessing.** We first circle a car three times while scanning the car with a smartphone for the attainment of RGB-D images and its point clouds. Then we use Colmap (Schonberger & Frahm, 2016) and SAM (Kirillov et al., 2023) to obtain poses and remove the background point clouds. Finally, we use the 3DGS (Kerbl et al., 2023b) trained on the processed data to obtain 3D car model.

**3D Generation with Diffusion Prior.** Some current works (Jun & Nichol, 2023; Nichol et al., 2022) leverage a 3D diffusion model to learn the representation of 3D structure. However, these methods lack generalization ability due to the scarcity of 3D data. To facilitate 3D generation without direct supervision of 3D data, image or multi-view diffusion models are often used to guide the 3D creation process. Notable approaches like DreamFusion (Poole et al., 2022b) and subsequent works (Metzer et al., 2023; Lin et al., 2023) use an existing image diffusion model as a scoring function, applying Score Distillation Sampling SDS loss to generate 3D objects from textual descriptions. These methods, however, suffer from issues such as the Janus problem (Poole et al., 2022b; Metzer et al., 2023) and overly saturated textures. Inspired by Zero123 (Liu et al., 2023c), several recent works (Stability.AI, 2023; Shi et al., 2023a; Liu et al., 2023e; Kong et al., 2024; Zheng & Vedaldi, 2023; Melas-Kyriazi et al., 2024; Liu et al., 2023b;a; Qian et al., 2023) refine image or video diffusion models to better guide the 3D generation by producing more reliable multi-view images. However, these generative methods fail to generate high-quality cars due to the lack of the prior of real cars.

## 3 PROPOSED 3DREALCAR DATASET

### 3.1 DATA COLLECTION AND ANNOTATION

As shown in Figure 4, our dataset is collected using smartphones, specifically iPhone 14 models, adopting ARKit APIs (Apple, 2021) to scan cars for their point clouds and RGB-D images. The data collection process is conducted under three distinct lighting conditions, such as standard, reflective, and dark. These lighting conditions represent the lighting states of vehicle surfaces. It is important to note that all data collection is performed with the consent of owners. During the scanning process, the car should be stationary while we meticulously circle the car three times to capture as many views as possible. For each loop, we adjust the height of the smartphone to obtain images from different angles. Furthermore, we try our best to make sure captured images contain the entire car body without truncation. To preserve the privacy of owners, we make license plates and other private information obfuscated. To construct a high-quality dataset, we filter out some instances with blurred, out-of-focus, and occluded images. We also provide detailed annotations for car brands, types, and colors. Particularly, we provide the car parsing maps for each car with thirteen classes in our dataset as shown in Figure 1 for the advancement of car component understanding tasks.

### 3.2 DATA PREPROCESSING

**Background Removal.** Since we only reconstruct cars for the 3D car reconstruction task, the background should be removed. Recent Segment Anything Model (SAM) (Kirillov et al., 2023) demonstrates powerful context recognition and segmentation performance. However, SAM needs a bounding box, text, or point as a driving factor for accurate segmentation. Therefore, we employ Grounding DINO (Liu et al., 2023d) as a text-driven detector with a detection prompt with "car" for

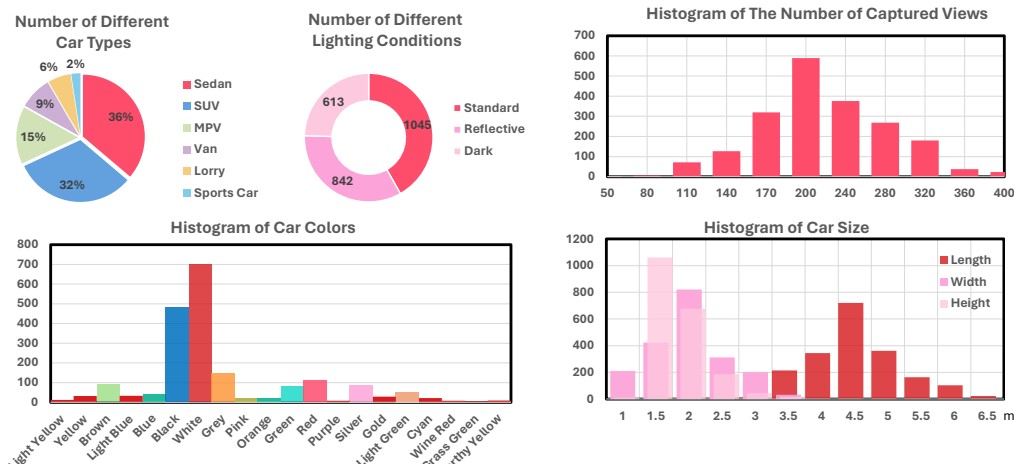

Figure 5: **The distributions of our 3DRealCar dataset.** We show distributions of car types, lighting conditions, captured views, car colors, and car size. We try our best to capture cars with various colors and types for the diversity of our dataset.

the attainment of car bounding boxes. With these bounding boxes, we use SAM to obtain the masks from captured images. The point cloud initialization is demonstrated useful for the convergence of 3D Gaussian Splatting (Kerbl et al., 2023b). Except for the removal of the background in 2D images, we still need to remove the background point clouds. Therefore, we first project the 3D point clouds into 2D space with camera parameters. Then, we can eliminate background point clouds with masks and save them for further processing.

**Orientation Rectification.** As shown in Figure 4, we utilize Colmap (Schonberger & Frahm, 2016) to reconstruct more dense point clouds and obtain accurate camera poses and intrinsics because we find that the estimated poses by the smartphone are not accurate. However, after the removal of the background point clouds, we find that the car orientation of the point cloud is random, which leads to the subsequent render task being uncontrollable. Given camera poses $P = \{p_i\}_1^{\mathcal{N}}$, where $\mathcal{N}$ is the number of poses, we use Principal Component Analysis (PCA) (Abdi & Williams, 2010) to obtain a PCA component $\mathcal{T} \in \mathbb{R}^{3 \times 3}$. The PCA component is the principal axis of the data in 3D space, which represents rotation angles to each axis. Therefore, we leverage it to rectify the postures of cars parallel to the x-axis. However, this process cannot guarantee cars facing along the x-axis. Therefore, in some failure cases, we manually interfere and adjust the orientation along the x-axis. With the fixed car orientation, we can control rendered poses for the subsequent tasks.

**Point Cloud Rescaling.** The size of the point clouds reconstructed by Colmap (Schonberger & Frahm, 2016) does not match the real-world size, which inhibits the reconstruction of a practically sized 3D car. To address this, we calculate the bounding box of the scanned foreground point clouds to obtain its actual size in the real-world scene. Then, we rescale the rectified point clouds into the real size. In addition to the rescaling of the point clouds, we also need to adjust the camera poses. We rescale translations of camera poses using a scale factor calculated by the ratio of scanned point cloud size and Colmap point cloud size. After these rescaling processes, we use rescaled point clouds to reconstruct a 3D car model through recent state-of-the-art methods, like 3DGS (Kerbl et al., 2023b).

### 3.3 DATA STATISTICS

In our 3DRealCar, we provide detailed annotations for researchers to leverage our dataset for different tasks. During the data annotating, we discard the data with the number of views less than fifty. As we can observe in Figure 1 and 2, we collect our dataset under real-world scenes and meticulously scan dense views. Therefore, cars in our dataset possess dense views and realistic texture, which is necessary for the application in a real-world setting.

As shown in Figure 5, we conduct detailed statistical analysis to show the features of our dataset. Our dataset mainly contains six different car types, such as Sedan, SUV, MPV, Van, Lorry, and Sports Car.

Among them, sedans and SUVs are common to collect in real life, so their volume dominates in our dataset. We also count the number of different lighting conditions on cars. The standard condition means the car is well-lit and without strong specular highlights. The reflective condition means the car has strong specular highlights. Glossy materials bring huge challenges to recent 3D reconstruction methods. The dark condition means the car is captured in an underground parking so not well-lit. To promote high-quality reconstruction, we save the captured images in high resolution (1920×1440) and also capture as many views as possible. The number of captured images per car is an average of 200. The number of views ranges from 50 to 400. To enrich the diversity of our dataset, we try our best to collect as many different colors as possible. Therefore, our dataset contains more than twenty colors, but the white and black colors still take up most of our dataset. In addition, we also show the distribution of car size, in terms of their length, width, and height. We obtain their sizes by computing the bounding boxes of the scanned point clouds. Thanks to different car types, the sizes of cars are also diverse.

# 4 OVERVIEW OF 3DREALCAR TASKS

## 4.1 2D TASKS

**Corner-case scene 2D Detection** (Ultralytics, 2023; Zhang et al., 2022a; Zong et al., 2023): Given a serial of images $I = \{I_i\}_1^{\mathcal{N}}$, this task aims to detect vehicles as accurately as possible. However, in some corner cases, like car accidents, detectors sometimes fail to detect target vehicles since this kind of scene is rare or not in the training set. Therefore, this task has crucial significance in building a reliable self-driving system, especially for accident scenarios.

**2D Car Parsing** (Chen et al., 2017; Hong et al., 2021; Xie et al., 2021; Kirillov et al., 2020): Given a serial of images $I = \{I_i\}_1^{\mathcal{N}}$, this task aims to segment car parsing maps $S = \{S_i\}_1^{\mathcal{N}}$. With annotated parsing maps, we can train a model to understand and segment each component of cars. This task can assist self-driving systems with more precise recognition.

## 4.2 3D TASKS

**Neural Field-based Novel View Synthesis** (Müller et al., 2022; Kerbl et al., 2023b; Huang et al., 2024): Given a serial of images $I = \{I_i\}_1^{\mathcal{N}}$ and matched poses $P = \{p_i\}_1^{\mathcal{N}}$, where $\mathcal{N}$ is the number of images and poses, the task of Neural Field-based Novel View Synthesis aims to reconstruct Neural Field model of a object or a scene. The reconstructed model is usually used to render 2D images with different views for the evaluation of the performance of novel view synthesis.

**Diffusion-based Novel View Synthesis** (Liu et al., 2023c;e; Stability.AI, 2023): Given a serial of reference images $I^{ref} = \{I_i^{ref}\}_1^{\mathcal{N}}$, reference poses $P^{ref} = \{p_i^{ref}\}_1^{\mathcal{N}}$, target images $I^{target} = \{I_i^{target}\}_1^{\mathcal{N}}$, and target poses $P^{target} = \{p_i^{target}\}_1^{\mathcal{N}}$, recent 3D generative models, such as Zero123 (Liu et al., 2023c), Syncdreamer (Liu et al., 2023e), and Stable-Zero123 (Stability.AI, 2023), take relative poses and reference images as inputs and generate target images. However, these models cannot generalize well to real car objects since they are trained on large-scale synthetic datasets (Deitke et al., 2023; 2024). In this work, we will demonstrate that our dataset can improve the robustness of these generative models to real cars.

**Single Image to 3D Generation** (Poole et al., 2022a; Sun et al., 2023; Tang et al., 2023): Given a text prompt or single image, recent 3D generation methods generate 3D objects with Score Distillation Sampling (SDS) (Poole et al., 2022a) and diffusion generative models (Rombach et al., 2022; Liu et al., 2023c; Stability.AI, 2023). However, these methods cannot generate high-quality 3D cars due to the lack of the prior of real cars in 3D-based diffusion models. Therefore, we would demonstrate the value of our dataset by improving recent 3D generation for real cars.

# 5 EXPERIMENTS

## 5.1 EXPERIMENTAL SETUP

**Corner-case 2D Detection.** In this task, we leverage the reconstructed cars to simulate rare and corner-case scenes. To be specific, we use Nuscenes (Caesar et al., 2020) as background to

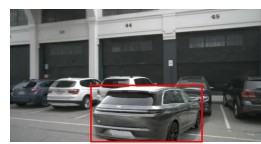 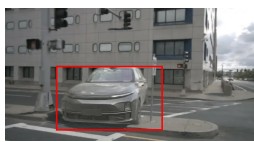 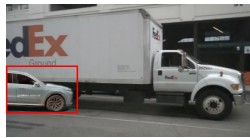

Figure 6: **The simulated corner-case scenes.** These scenes are rare but very important in real life. We use a red rectangle to highlight the simulated vehicles. These corner-case scenes show some vehicles have potential risks to traffic safety.

Table 2: **Detection improvements by simulated data for corner-case scenes.** We leverage lightweight YOLO serials models and recent state-of-the-art models for evaluation. We report the metric by calculating mAP@0.5 on the CODA dataset Li et al. (2022).

| Simulated Data | YOLOv5n | YOLOv5s | YOLOv8n | YOLOv8s | DINO | CO-DETR |
|---|---|---|---|---|---|---|
| 1000 | 0.285 | 0.341 | 0.299 | 0.371 | 0.437 | 0.465 |
| 2000 | 0.304 | 0.357 | 0.312 | 0.366 | 0.452 | 0.481 |
| 3000 | 0.345 | 0.389 | 0.357 | 0.403 | 0.495 | 0.517 |
| 4000 | 0.357 | 0.408 | 0.386 | 0.413 | 0.543 | 0.551 |
| 5000 | **0.361** | **0.426** | **0.386** | **0.435** | **0.571** | **0.582** |

simulate corner-case scenes with reconstructed cars and leverage recent popular detectors, like YOLOv8 (Ultralytics, 2023), as detectors for evaluation. To evaluate the robustness of detectors in corner-case scenes, we use the test part of the corner-case dataset, CODA (Li et al., 2022) as a testing set. Since we focus on the corner-case scenes of cars, so we only evaluate a car class.

**2D Car Parsing.** In this task, we utilize DeepLabV3(Chen et al., 2017), DDRNet (Hong et al., 2021), SegFormer (Xie et al., 2021), and PointRend (Kirillov et al., 2020) to benchmark our dataset. To be specific, we split 80% of our car parsing maps in 3DRealCar as the training set and the rest of 20% as the testing set.

**Neural Field-based Novel View Synthesis.** In this task, we randomly choose 100 instances from each lighting condition in our dataset and split 80% of the views per instance as the training set and the rest of 20% as the testing set. Specifically, we employ recent state-of-the-art neural field methods, including Instant-NGP (Müller et al., 2022), 3DGS (Kerbl et al., 2023b), GaussianShader (Jiang et al., 2023), and 2DGS (Huang et al., 2024) to benchmark our dataset.

**Diffusion-based Novel View Synthesis.** We finetune Zero123-XL (Liu et al., 2023c) on our 3DReal-Car dataset to enhance its generalization to real cars. Note that since the training of diffusion-based models needs entire objects centered on images, we use the images rendered by our trained 3D models as training images.

**Single Image to 3D Generation.** In this task, we exploit Dreamcraft3D (Sun et al., 2023) as our baseline. Dreamcraft3D exploits Stable-Zero123 (Stability.AI, 2023) as a prior source for providing 3D generative prior. By fine-tuning Stable-Zero123 on our dataset, we enable it to obtain car-specific prior so it generalizes well to real cars.

### 5.2 2D TASKS

**Corner-case 2D Detection.** As shown in Table 2, we employ YOLOv5 and YOLOv8 serial models, DINO (Zhang et al., 2022a), and CO-DETR Zong et al. (2023) as our detectors for evaluation. To evaluate the performance of models in corner-case scenes, we leverage the test part of the CODA dataset (Li et al., 2022) as our testing set. In particular, when we increase the training simulated data from 500 to 5,000, the performance of detectors is also improved by a large margin. This phenomenon demonstrates that our simulated data is effective in improving a detector robust to corner-case scenes. We provide the visualizations of simulated corner-case scenes in Figure 6. The detailed simulation process and more visualizations can be seen in the supplementary.

**2D Car Parsing.** We conduct benchmarks for car parsing maps of our dataset using recent image segmentation methods, such as DeepLabV3(Chen et al., 2017), PointRend(Kirillov et al., 2020),

Table 3: **Benchmark results on 2D car parsing of our 3DRealCar dataset.** We use recent advanced image segmentation methods Chen et al. (2017); Hong et al. (2021); Xie et al. (2021); Kirillov et al. (2020) to benchmark our dataset.

| Method | DeepLabV3 | PointRend | DDRNet | SegFormer |
|---|---|---|---|---|
| mIOU ↑ | 0.556 | 0.562 | 0.603 | **0.613** |
| mAcc ↑ | 0.616 | 0.619 | 0.659 | **0.663** |

Table 4: **Quantitative comparisons of SOTA 3D Generation method, Dreamcraft3D Sun et al. (2023) and its improved version by trained on our dataset.** CD denotes Chamfer Distance.

| Method | CLIP-I ↑ | Hausdorff ↓ | CD ↓ |
|---|---|---|---|
| Dreamcraft3D | 0.812 | 1.572 | 0.587 |
| +our dataset | **0.847** | **1.364** | **0.371** |

Table 5: **Benchmark results on 3D reconstruction of our 3DRealCar dataset.** We present the 3D reconstruction performance of recent state-of-the-art methods in three lighting conditions, standard, reflective, and dark, respectively. The best results are highlighted.

| Method | Standard | | | Reflective | | | Dark | | |
|---|---|---|---|---|---|---|---|---|---|
| | PSNR ↑ | SSIM↑ | LPIPS↓ | PSNR↑ | SSIM↑ | LPIPS↓ | PSNR↑ | SSIM↑ | LPIPS↓ |
| Instant-NGP Müller et al. (2022) | 27.31 | 0.9315 | 0.1264 | 24.37 | 0.8613 | 0.1962 | 23.17 | 0.9152 | 0.1642 |
| 3DGS Kerbl et al. (2023b) | 27.47 | **0.9367** | **0.1001** | 24.58 | 0.8647 | 0.1852 | **23.51** | **0.9181** | **0.1613** |
| GaussianShader Jiang et al. (2023) | **27.53** | 0.9311 | 0.1109 | **25.41** | **0.8684** | **0.1423** | 23.39 | 0.9172 | 0.1631 |
| 2DGS Huang et al. (2024) | 27.34 | 0.9341 | 0.1095 | 23.19 | 0.8509 | 0.2041 | 22.63 | 0.9148 | 0.1681 |

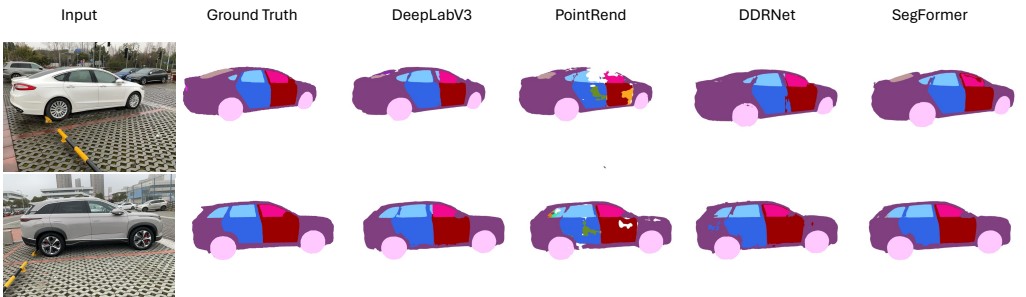

Figure 7: **Qualitative comparisons among recent advanced image segmentation methods.** We select the inputs from the testing set of our images and evaluate the capacity of car component understanding for each method.

DDRNet(Hong et al., 2021), and SegFormer(Xie et al., 2021). The quantitative performance for these methods on our dataset is summarized in Table 3. Visual comparisons are provided in Figure 7. Our high-quality dataset enables these methods to achieve promising performance, highlighting its potential for application in self-driving systems. In particular, our car parsing annotations encourage self-driving systems to recognize different components of cars in practical scenarios for safer automatic decisions.

## 5.3 3D TASKS

**Neural Field-based Novel View Synthesis.** As depicted in Table 5, we show benchmark results of recent state-of-the-art neural field methods, such as Instant-NGP (Müller et al., 2022), 3DGS (Kerbl et al., 2023b), GaussianShader (Jiang et al., 2023), and 2DGS (Huang et al., 2024) on our dataset. To the standard lighting condition, we can find that recent methods are capable of achieving PSNR more than 27 dB, which means these methods can reconstruct relatively high-quality 3D cars from our dataset. However, the reflective and dark condition results are lower than the standard. These two parts of our 3DRealCar bring two challenges to recent 3D methods. The first challenge is the reconstruction of specular highlights. Due to the particular property of cars, materials of car surfaces are generally glossy, which means it would produce plenty of specular highlights if cars are exposed to the sun or strong light. The second challenge is the reconstruction in a dark environment. The training images captured in the dark environment lose plenty of details for reconstruction. Therefore, how to achieve high-quality reconstruction results from these two extreme lighting conditions is a challenge to recent methods. 3D visualizations can be found on our project page.

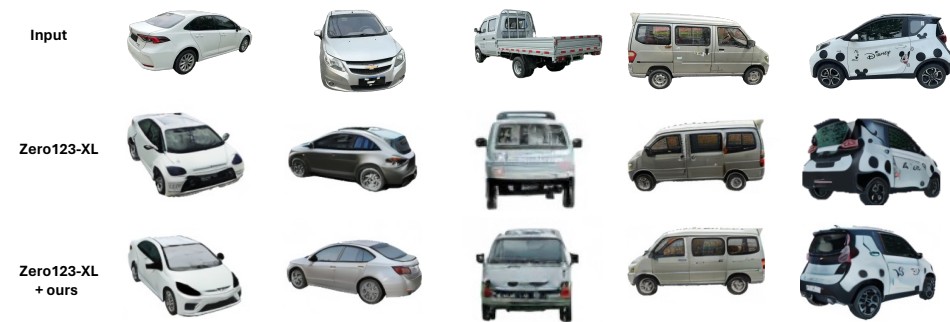

Figure 8: **Visualizations of diffusion-based novel view synthesis.** we compare the results of the recent state-of-the-art diffusion-based method, Zero123-XL (Liu et al., 2023c) and its improvement by training on our dataset. Our dataset provides car-specific prior for the generative model to generate more photorealistic car images.

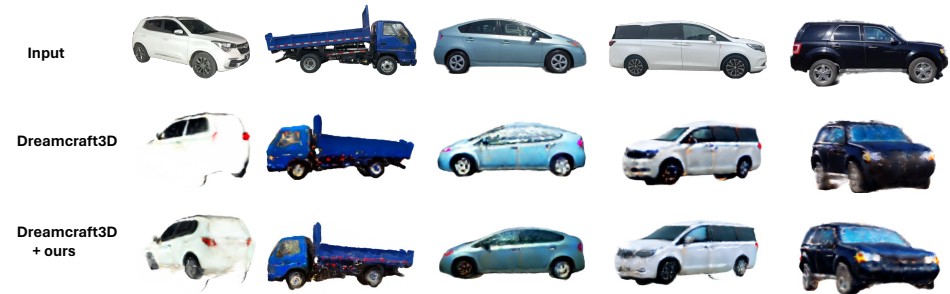

Figure 9: **Visualizations of single-image-to-3D generation.** we compare the results of the recent state-of-the-art single-image-to-3D method, Dreamcraft3D (Sun et al., 2023) and is enhanced version by training on our dataset.

**Diffusion-based Novel View Synthesis.** As illustrated in Figure 8, we show visual comparisons of Zero123-XL (Liu et al., 2023c) and our improved version by training on our dataset. As we can see, given input images, we use Zero123-XL and our improved version to synthesize novel views. In this figure, we can find that Zero123-XL prefers to generate synthetic results with unrealistic texture and geometry, due to the lack of prior for real objects. In contrast, our improved version of Zero123-XL can generate photorealistic geometry and texture, which demonstrates the effectiveness of our dataset.

**Single Image to 3D Generation.** As depicted in Figure 9, we visualize 3D generation results of the recent state-of-the-art single-image-to-3D method, Dreamcraft3D (Sun et al., 2023), along with its improved version by our dataset. This figure shows that Dreamcraft3D sometimes fails to generate complete geometry or realistic texture, due to the scarcity of the real car prior. As shown in Table 4, we also show quantitative comparisons of Dreamcraft3D and its improved version. CLIP-I means the similarity of rendered images with the original input. The quantitative and qualitative results indicate our dataset significantly improves 3D generation performance typically in terms of geometry and texture. These results underscore the effectiveness of our 3DRealCar dataset.

## 6 CONCLUSION

In this paper, we propose the first large-scale high-quality 3D real car dataset, named 3DRealCar. The collected dense and high-resolution 360-degree views for each car can be used to reconstruct a high-quality 3D car. Extensive experiments demonstrate the efficacy and challenges of our 3DRealCar in 3D reconstruction. Thanks to the reconstructed high-quality 3D cars from our dataset and car-part level annotations, our dataset can be utilized to support various tasks related to cars. In addition, the benchmarking results can serve as baselines for prospective research. Although 3DRealCar currently only has car exterior views, we intend to provide both exterior and interior views in the future to further promote the reconstruction of more intact 3D cars.

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

## A    BROADER IMPACTS STATEMENT

The introduction of our 3DRealCar dataset has profound effects on self-driving research. We expect this dataset can encourage extensive research to promote the advancement of the community.

**Research Impacts.** By providing dense 360-degree views of cars with point clouds as initialization, our 3DRealCar can be used to reconstruct high-quality 3D real cars for 3D printing and simulation in corner-case scenes. By providing detailed car parsing map annotations, our dataset can be leveraged to segment 2D car components or point clouds. Note that our 3DRealCar is the first dataset providing 3D car parsing annotations. In our 3D reconstruction benchmarking experiments, the reflective and dark lighting conditions of our dataset bring challenges to existing methods to reconstruct 3D cars under awful lighting conditions. We expect our dataset to encourage widespread collaboration and accelerate the exploration of 3D real car reconstruction, parsing, and simulation.

**Societal Impacts.** We collect our 3DRealCar dataset with the consent of the owners. In addition, we blur license plates and other private information. We try our best to hide and preserve the privacy of owners. Therefore, our dataset would not have any privacy violation problems. Due to our dataset focusing on a car class, we believe our dataset has the potential to be employed in future self-driving research and improve self-driving systems further.

## B    LIMITATION AND DISCUSSION

Although our 3DRealCar is the largest dataset for the 3D real car dataset so far (2500 car instances with annotations), its scale is still limited compared to other datasets in the computer vision community. Therefore, we will further extend our dataset in the future. Moreover, our 3DRealCar dataset only provides the exterior views of cars without interior views. It is very crucial to reconstruct both exterior and interior views of cars for car marketing agencies. We will collect both exterior and interior views in the future to further extend our 3D real car dataset for intact 3D car models.

## C    EXPERIMENTAL SETTINGS

Note that all models used in this work are publicly available. Each model we use is linked below:

1. **3D Reconstruction:** Instant-NGP (Müller et al., 2022) ⚙, 3DGS (Kerbl et al., 2023b) ⚙, GaussianShader (Jiang et al., 2023) ⚙, and 2DGS (Huang et al., 2024) ⚙.

2. **2D Car Parsing:** MMsegmentation ⚙. This repository includes all 2D segmentation models (Chen et al., 2017; Hong et al., 2021; Xie et al., 2021; Kirillov et al., 2020) we used in this work.

3. **Novel View Synthesis:** Zero-123-XL (Liu et al., 2023c) ⚙.

4. **3D Generation:** DreamCraft3D (Sun et al., 2023) ⚙.

5. **Corner-case Simulation:** YOLOv5 and YOLOv8 (Ultralytics, 2023) ⚙, DINO (Zhang et al., 2022a) ⚙, CO-DETR (Zong et al., 2023)⚙, and libcom (Niu et al., 2021) ⚙. Specifically, we use YOLOv5 and YOLOv8 serial models, DINO, and CO-DETR as detectors and libcom for the simulation of corner-case scenes.

We express great appreciation to the authors of the aforementioned repositories for their invaluable contributions. For the GPU specification, we use 8 A100 GPUs for 3D reconstruction, 3D generation, and novel view synthesis. We utilize 2 3090 GPUs for other tasks. We use the default hyperparameters for training.

## D    DETAILED SIMULATION PROCESS AND ADDITIONAL VISUALIZATIONS

In this section, we show how we simulate corner-case scenes. As shown in Figure 10, we use images from Nuscenes (Caesar et al., 2020) as backgrounds and leverage ViT-Adapter (Chen et al., 2022c) to segment entire scenes for road masks. Then, we copy and paste the rendered images from the reconstructed high-quality 3D cars into the backgrounds with the guidance of road masks. In

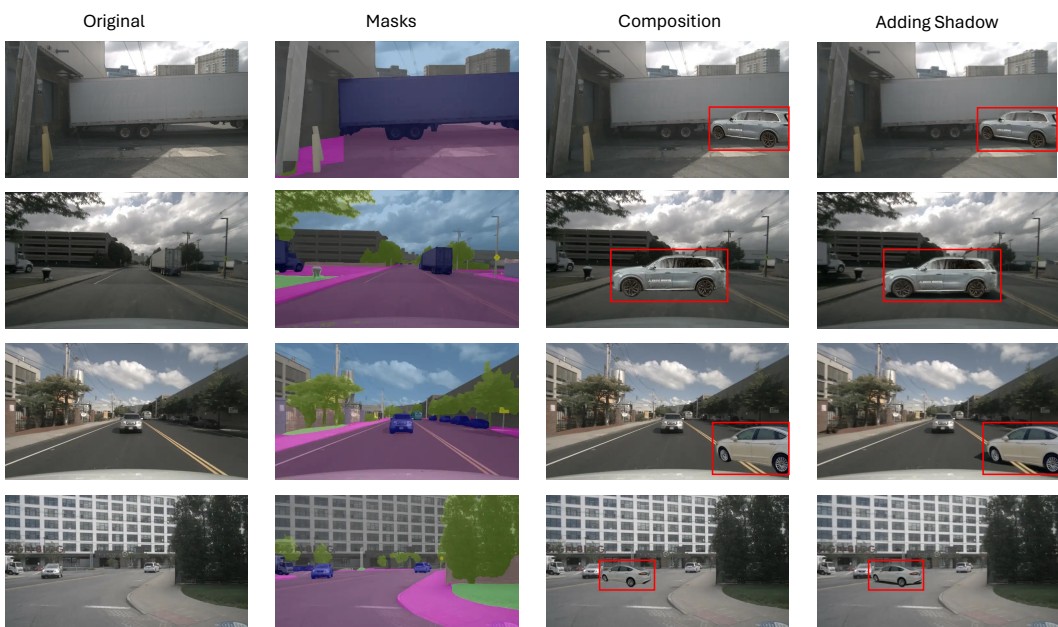

Figure 10: **Visualizations of ablating simulation procedures.** We use a red rectangle to highlight the simulated vehicles.

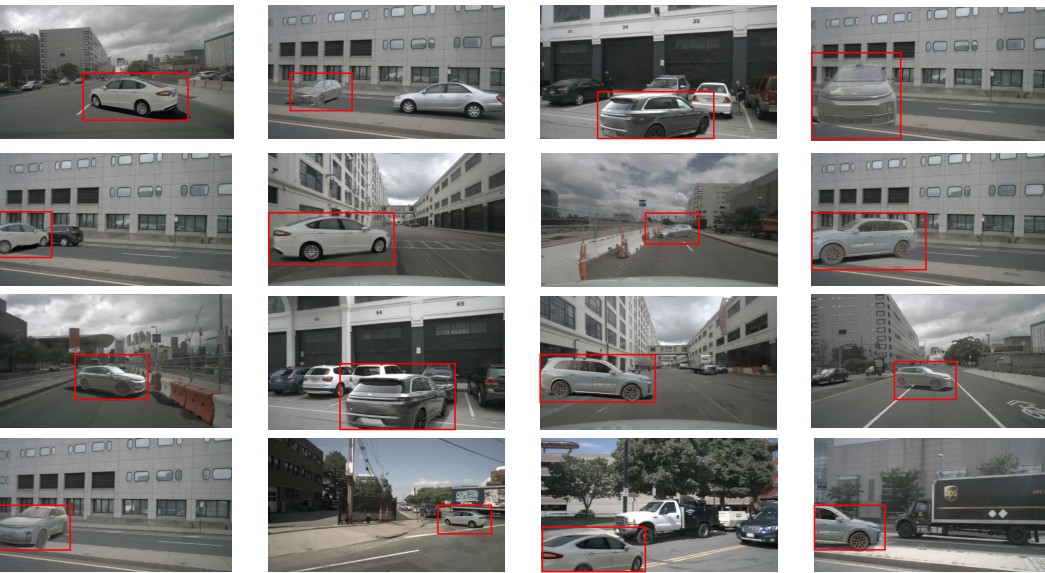

Figure 11: **More visualizations of simulated corner-case scenes.** We use a red rectangle to highlight simulated vehicles. These corner-case scenes show some vehicles have potential risks to traffic safety.

particular, we blur the edge between simulated foregrounds and backgrounds and then we use a color transfer algorithm (Reinhard et al., 2001) to make the whole simulated scene look harmonious. Finally, we use the shadow generation method in libcom (Niu et al., 2021) to add shadow for the simulated cars such that the entire scene looks photorealistic. However, this simulation method would generate some unreasonable scenes. Therefore, we manually intervene to select photorealistic corner-case scenes. Additional simulation results are shown in Figure 11.

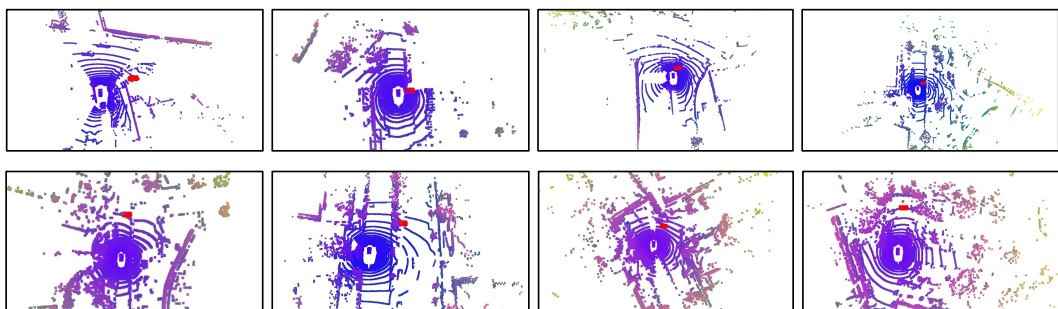

Figure 12: **Visualizations of point cloud inserting.** We use red color to annotate inserting vehicular point clouds with high density for better differentiation.

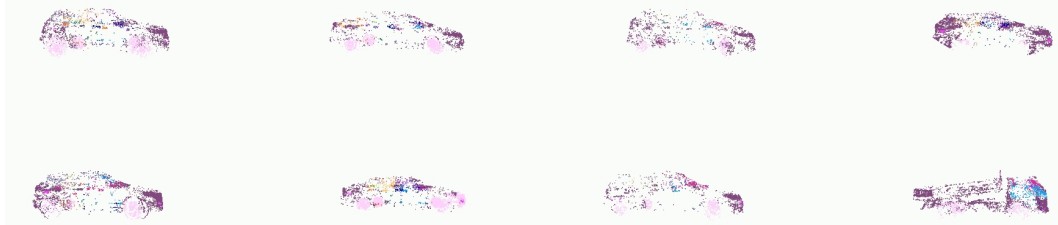

Figure 13: **Visualizations of 3D point cloud parsing.** With 2D car parsing map annotations, we lift the 2D car parsing maps into 3D point clouds and segment car components.

## E    SIMULATED LIDAR SCENES

As depicted in Figure 12, we can insert our car point clouds into lidar scenes to simulate corner-case scenes, like a car passing or parking horizontally in front of the ego car. To better differentiate the inserted cars, we set them with dense point clouds and red color. In a practical scene, the vehicular point clouds should be sparse and only have one size that could be scanned by the lidar. Therefore, when we apply the inserted vehicular point clouds into a scene, we should make the vehicular point clouds sparse and only contain one side. By training on a variety of simulated scenarios, including rare or dangerous situations that are difficult to collect in real life, the self-driving system can learn to handle unexpected events more effectively.

## F    3D CAR PARSING

As shown in Figure 13, our dataset is the first to provide 3D car parsing annotations for parsing car components in 3D space. Thanks to that we provide 2D car parsing maps for every instance in our 3DRealCar dataset, we can lift 2D parsing maps to 3D and segment each component for point clouds and meshes. The primary purpose of these 3D car parsing maps is to enable precise and comprehensive analysis of vehicle structures, which is crucial for applications such as autonomous driving, vehicle design, vehicle editing, and virtual reality simulations. By using these detailed 3D parsing maps, developers and researchers can improve object recognition algorithms and enhance collision detection systems. Furthermore, this dataset facilitates the training of machine learning models to better understand the spatial relationships and physical attributes of car components, leading to more advanced and reliable automotive technologies.

## G    CONSENT FORM FOR 3DREALCAR

Since our dataset includes license plate information, we obtain consent from participants and require them to sign the consent form shown in Figure 14 before data collection. We ensure that no personally identifiable information, like the plate number, would appear in our published dataset. Additionally, it is crucial to note that our dataset is intended solely for academic use and is not permitted for commercial purposes.

---

# Consent Form for Collection of 3D Real Car Dataset

**Purpose:**
The purpose of this research is to create a 3D real car dataset for the development of various automotive and machine learning applications. The data will be used to improve technologies such as autonomous driving, vehicle detection, and augmented reality.

**Procedures:**
Participants will provide cars for data collection where cars will be scanned using 3D imaging technology. This collection will take place at designated locations and last approximately several minutes. Participants' vehicles will be scanned from multiple angles to create a comprehensive dataset.

**Risks:**
There are minimal risks associated with participation in this study. Participants will not be exposed to any hazardous conditions, and the 3D scanning technology is non-invasive.

**Benefits:**
Participants will contribute to the advancement of automotive technology, potentially leading to safer and more efficient vehicle systems. Participants will be paid at a rate of $ 200 per hour.

**Confidentiality:**
Identifying information will be removed from the dataset. Only authorized personnel will have access to the raw data. We will hide any private information, like the plate number.

**Consent:**
I, the undersigned, consent to participate in the recording of a 3D real car dataset. I understand the purpose of the study, the procedures involved, the risks, and the benefits. I understand that my participation is voluntary and that I can withdraw at any time without penalty.

Date: _______________________________
Name of Participant: _______________________________
Signature of Participant: _______________________________
Name of Researcher: _______________________________
Signature of Researcher: _______________________________

Figure 14: Consent Form for Collection of 3DRealCar

