# OpenReview forum: "3DRealCar: An In-the-wild RGB-D Car Dataset with 360-degree Views"
_ICLR.cc/2025/Conference — Submitted to ICLR 2025_

### Official Review · Reviewer_fi6m · 2024-10-15

**Soundness:** 3
**Presentation:** 3
**Contribution:** 2
**Rating:** 5
**Confidence:** 4

**Summary:**

The paper proposes 3DRealCar, a large-scale 3D real car dataset with RGB-D images and point clouds of 2,500 cars captured in real-world environments. It addresses the limitations of existing car datasets that usually use synthetic data or have low-quality data. This dataset includes cars under three lighting conditions (standard, reflective, and dark), promoting research in 3D car reconstruction, parsing, and novel view synthesis. Benchmarks with state-of-the-art methods demonstrate that existing methods struggle in reflective and dark lighting conditions, emphasizing the dataset’s value for improving 3D reconstruction methods.

**Strengths:**

1. This paper addresses the limitations of current car datasets by capturing real-world car data with diverse samples, high-quality images, and point clouds.
2. The idea of introducing 3 different lighting conditions is interesting and makes up for the shortcomings of existing datasets
3. The extensive experiments demonstrate the effectiveness of this dataset on various tasks.

**Weaknesses:**

1. Though collecting data is costly, the contribution of this work is only on the dataset and lacks technical contribution for 3D/2D car understanding.
2. This dataset captures cars under three lighting conditions, including reflective, standard, and dark; however, it seems to lack data of the same car under all three conditions, which limits the exploration of the effects of lighting on car appearance.

**Questions:**

Data of the same car under different lighting conditions could be added to enhance the analysis of lighting effects. Additionally, including metadata such as environmental maps or more detailed car materials could improve the dataset's versatility.

---

> ### Author Response · Authors · 2024-11-20
> **Response to Reviewer fi6m**
>
> We sincerely thank Reviewer fi6m for the appreciative and constructive comments. We appreciate that Reviewer fi6m recognizes the strengths of our paper, including the proposed high-quality dataset, various environmental lighting, and comprehensive experiments.  We address the concerns raised by Reviewer fi6m below.
>
>
> **Q1.** The contribution of this work is only on the dataset and lacks technical contribution.
>
> **A1.** Here, we provide quite valuable works with references [1-5] that only contain the dataset and benchmark contribution and are all accepted by ICLR, which all benefit the community a lot. Therefore, ICLR accepted papers that only have the dataset and benchmark contribution. In addition, on the “Call for Paper” of the ICLR official website, datasets and benchmarks are clearly included in the scope of the ICLR submissions. So that both ICLR and the community encourage high-quality dataset works to get published, which take effort to get collected, organized, well-processed, and delivered to the community.
> Therefore, we believe that our paper that contains the dataset and benchmark contributions should be suitable as a potential manuscript that can be considered to be accepted by ICLR25.
>
>
>
> [1] Zheng L, Chiang W L, Sheng Y, et al. LMSYS-Chat-1M: A Large-Scale Real-World LLM Conversation Dataset[C]//The Twelfth International Conference on Learning Representations.
>
> [2] Sun X, Yao Y, Wang S, et al. Alice Benchmarks: Connecting Real World Re-Identification with the Synthetic[C]//The Twelfth International Conference on Learning Representations.
>
> [3] Jiang X, Zhao Y, Lin Y, et al. CircuitNet 2.0: An Advanced Dataset for Promoting Machine Learning Innovations in Realistic Chip Design Environment[C]//The Twelfth International Conference on Learning Representations. 2024.
>
> [4] Wang Q, Zhang Z, Liu Z, et al. EX-Graph: A Pioneering Dataset Bridging Ethereum and X[C]//The Twelfth International Conference on Learning Representations.
>
> [5] Zhu Y, Hwang J, Adams K, et al. Learning Over Molecular Conformer Ensembles: Datasets and Benchmarks[C]//The Twelfth International Conference on Learning Representations. 2023.
>
>
> **Q2:** This dataset seems to lack data of the same car under all three conditions.
>
> **A2:** Thank you for your constructive suggestion. This idea would help us to construct a more diverse and comprehensive dataset. In our dataset, we have the same car of the same car type under different lighting, which can be seen as a real-world relighting dataset for lighting decoupling research.  For example, our dataset has BMW 530Li cars with the same white color and different lighting conditions such as standard, reflective, and dark. We provide its Google Drive here: https://drive.google.com/file/d/1BP9BQQ7j-jvCzOIPXIvVJijl91qdm_jB/view?usp=sharing  Thank you for this idea again.
>
>
>
> **Q3:** Including metadata such as environmental maps or more detailed car materials could improve the dataset's versatility.
>
> **A3:** Thank you for your great insight.  We present some samples of the reconstructed meshes and material decompositions here.
> https://drive.google.com/file/d/11xU5wWemcc1g5OjZgdit3dNKIZiEyZKH/view?usp=sharing
> We will release the meshes, environmental maps, and material decompositions in our dataset to the public and increase our dataset's versatility.

---

> ### Author Response · Authors · 2024-11-25
> **Response to Reviewer fi6m**
>
> Dear Reviewer fi6m,
>
> We hope this message finds you well.
>
> We would like to ask if we have sufficiently addressed the concerns you previously raised. If there are still concerns requiring further clarification or elaboration, please kindly let us know, and we will be more than happy to provide additional details to ensure that all your concerns are fully addressed.
>
> Thank you for your valuable time and feedback.
>
>
> Best Regard
>
> 3DRealCar Authors

---

### Official Review · Reviewer_tJmk · 2024-11-04

**Soundness:** 2
**Presentation:** 3
**Contribution:** 1
**Rating:** 5
**Confidence:** 5

**Summary:**

This paper introduces a comprehensive 3D car dataset featuring high-quality 360-degree scans. The dataset comprises 2,500 cars, each scanned with 3D scanners to produce 200 high-resolution RGB-D views. Additionally, the dataset includes variations across three different lighting conditions: reflective, standard, and dark.

The authors detail the data collection methodology extensively. Data capture involved the use of RGB-D sensors, followed by the recovery of camera poses using Colmap Structure from Motion (SfM), and mask extraction utilizing Grounding-DINO. The 3D models were then reconstructed with 3DGS.

The paper discusses extensive 2D and 3D downstream experiments conducted using the dataset, which include 2D detection and segmentation, depth estimation, point cloud completion, and 3D car generation. The experiment results demostrate the effectiveness of the collected data on various tasks.

**Strengths:**

1. The structure of the paper is well-organized and the content is clearly presented;
2. The effectiveness of the dataset is demonstrated through its application to various downstream tasks, including various 2D and 3D tasks.
3. This paper introduces a novel 360-degree real car dataset. Previous literature has not extensively covered. Both the quality and the diversity of the dataset are commendable.

**Weaknesses:**

1. The data collection and processing method is not novel and is a standard approach to reconstructing 3D assets.
2. The paper does not highlight its uniqueness and irreplaceability, especially in terms of improving 2D parsing and 2D detection performance.
3. The results of the NVS task (Fig.8 and 9) are considerably inferior to the state of the art.
4. Many previous papers, including CADSim (https://arxiv.org/pdf/2311.01447), GeoSim (Chen et al., CVPR'21), and other related works, have demonstrated the capability to reconstruct cars from real-captured data, which I believe is a more extensible way to reconstruct car assets. The authors claim that their data is of higher quality, but the paper does not demonstrate the necessity of such high quality, especially for downstream tasks.

Although the paper provides a very useful dataset and makes a significant contribution, the downstream tasks and data collection processes are not novel and do not meet the acceptance standards of ICLR.

**Questions:**

See weakness.

---

> ### Author Response · Authors · 2024-11-20
> **Response to Reviewer tJmk**
>
> We sincerely thank Reviewer tJmk for the valuable comments. We appreciate that Reviewer tJmk recognizes our proposed dataset is very useful and makes a significant contribution, including various tasks in 2D and 3D downstream applications. We address the concerns raised by Reviewer tJmk below.
>
> **Q1:** The data collection and processing method is not novel and is a standard approach to reconstructing 3D assets.
>
> **A1:** In our contribution, we do not claim we propose a novel data collection method or processing method. Our contribution is the dataset itself. Different from other car datasets as shown in Table 1 and Figure 2, our dataset is a real car dataset and provides depth information, point cloud, car component annotations, and lighting conditions.  The different lighting conditions of our dataset from real-world lighting can be seen as a benchmark for existing 3DGS methods to reconstruct more realistic 3D objects.   Moreover, we propose detailed annotations and part-level segmentation maps for various tasks.  We focus on constructing a high-quality and comprehensive 3D real car dataset for a more reliable and safe self-driving system.
>
>
> **Q2:** The paper does not highlight its uniqueness and irreplaceability, especially in terms of improving 2D parsing and 2D detection performance.
>
> **A2:** As shown in Table 1 and Figure 2, we conduct comprehensive comparisons between our proposed dataset and existing 3D car datasets in terms of data volume, resolution, lighting condition, and quality. Note that we focus on the construction of a high-quality 3D real car dataset for a reliable self-driving system not proposing a method to improve the parsing or detection performance. The presented 2D parsing and detection results demonstrate our dataset can be used to train a segmentation model to segment part components of cars and a detector to detect various cars in corner-case scenarios as shown in Figure 6. We particularly focus on corner-case scenes because existing detectors are trained on datasets containing plenty of safe scenes. With our car component and car type annotations, we expect the prospective self-driving system can identify the type of other cars and recognize the car components. For example, if the self-driving car detects there is a truck with enormous cargo on its back, the self-driving car should keep a safe distance from the truck. With our reconstructed high-quality 3D car assets, we can build a photorealistic simulator to synthesize potentially dangerous scenes for the training of self-driving systems.
>
>
> **Q3:** The results of the NVS task (Fig.8 and 9) are considerably inferior to the state-of-the-art.
>
> **A3:** The original Zero-123 and Dreamcraft3D generate low-quality results because these models lack car-specific prior. To mitigate this problem, a multi-view car dataset is necessary to provide car-specific prior for these models.  Figs. 8 and 9 demonstrate that our proposed dataset can enhance existing 3D diffusion-based models to generate better 3D cars or novel views.  Therefore, we demonstrate that Zero-123 and Dreamcraft3D trained on our dataset can generate better results.
>
>
> **Q4:** CADSim and GeoSim have demonstrated the capability to reconstruct cars. The authors need to demonstrate the necessity of their dataset.
>
> **A4:** These methods were noted before our submission. However, these methods cannot reconstruct high-quality 3D car assets. CADSim and GeoSim are all trained on low-quality datasets. For example, [1] displays the reconstructed results of GeoSim. They do not show 360-degree videos of reconstructed cars, indicating they can only reconstruct well on the one side of cars. CADSim is trained on the MVMC dataset, which we discussed in Table 1 and Figure 2. MVMC only contains around 10 views. It definitely cannot be used to reconstruct high-quality 3D car assets. In 3D reconstruction, more training views can provide more supervision for the training of 3D objects for better results. In contrast, our dataset provides around 200 views that can be used to reconstruct high-quality 3D cars. We provide video demos on the project page within the link in the abstract. Reviewer can watch the 360-degree display of our reconstructed 3D car assets in our project page.
>
>
> [1] https://tmux.top/publication/geosim/

---

> > ### Comment · Reviewer_tJmk · 2024-11-20
> >
> > Given that this article's contribution is confined to the dataset itself and does not provide many state-of-the-art results in downstream tasks, I recommend that the authors enhance the downstream tasks section. Therefore, I decide to maintain my decision as `weak rejection`.

---

> > > ### Author Response · Authors · 2024-11-20
> > > **How does a dataset provide state-of-the-art results?**
> > >
> > > We need to claim again that our contribution is a dataset, not a method. If a dataset can provide state-of-the-art results, it is not necessary to research new models, such as CNN and Transformer for ImageNet.

---

> > > > ### Comment · Reviewer_tJmk · 2024-11-24
> > > >
> > > > I should clarify that I did not expect the authors outperforms state-of-the-art methods, instead at least the paper should provide results trained with sota algorithms. Additionally, a more thorough and critical analysis of the dataset would be beneficial. It would be particularly helpful if the authors could highlight the unique aspects of the dataset that distinguish it in various downstream tasks. As a result, this paper does not meet the high acceptance standards  of the ICLR conference
> > > >
> > > > As for evaluation algorithms, I recommend the authors to provide more results using more recent algorithms. Most of the algorithms evaluated in this paper and out-dated. Regarding the NVS task, I suggest that the authors to check out with the recent advancements, for example, Zero123++ (which offers more stability and performance over Zero123), as well as other state-of-the-art methods such as InstantMesh, One-2-3-45++, SyncDreamer, etc. These results could provide valuable benchmarks results for the reader. For 3D reconstruction, I recommend exploring recent publications like PGSR, InstantNSR, and NeuralAngelo. These papers offer much more stable and robust reconstruction results. In the area of Semantic Segmentation, the methods employed appear to be outdated. I am wondering if this dataset could provide performance benefits given SOTA foundations models like SAM.
> > > >
> > > > Furthermore, I expect the authors to show more in-depth analysis thoughts about this dataset and benchmark. When the above concerns are carefully addressed, this paper would be a much better research work.

---

> ### Author Response · Authors · 2024-11-20
> **Dataset Contribution is Significant to Community**
>
> Here, we provide quite valuable works with references [1-5] that only contain the dataset and benchmark contribution and are all accepted by ICLR. These contributions have significantly benefited the scientific community. Consequently, it is evident that ICLR recognizes and accepts papers that offer substantial contributions in the form of datasets and benchmarks. Furthermore, the **Call for Papers** section on the ICLR official website explicitly includes datasets and benchmarks within the scope of acceptable submissions. This policy reflects ICLR's and the broader community's encouragement for the publication of high-quality dataset works, which require considerable effort to collect, organize, process, and present to the scientific community. Given this context, we are confident that our paper, which includes both dataset and benchmark contributions, aligns well with the criteria for potential acceptance at ICLR25.
>
> [1] Zheng L, Chiang W L, Sheng Y, et al. LMSYS-Chat-1M: A Large-Scale Real-World LLM Conversation Dataset[C]//The Twelfth International Conference on Learning Representations.
>
> [2] Sun X, Yao Y, Wang S, et al. Alice Benchmarks: Connecting Real World Re-Identification with the Synthetic[C]//The Twelfth International Conference on Learning Representations.
>
> [3] Jiang X, Zhao Y, Lin Y, et al. CircuitNet 2.0: An Advanced Dataset for Promoting Machine Learning Innovations in Realistic Chip Design Environment[C]//The Twelfth International Conference on Learning Representations. 2024.
>
> [4] Wang Q, Zhang Z, Liu Z, et al. EX-Graph: A Pioneering Dataset Bridging Ethereum and X[C]//The Twelfth International Conference on Learning Representations.
>
> [5] Zhu Y, Hwang J, Adams K, et al. Learning Over Molecular Conformer Ensembles: Datasets and Benchmarks[C]//The Twelfth International Conference on Learning Representations. 2023.

---

> ### Author Response · Authors · 2024-11-24
> **Resonse to  Reviewer tJmk (Unresonable Suggestion)**
>
> We appreciate your reply. You claim "Most of the algorithms evaluated in this paper and outdated". However, most methods we compare in 3D reconstruction and 3D generation are from 2023 and 2024. **Why do you let us compare some works not reasonable?** For example, InstantNSR focuses on **human modeling** and was published in 2022. Isn't it out-of-date? What is its relationship with our car dataset? You don't know our dataset focuses on cars?

---

> > ### Comment · Reviewer_tJmk · 2024-11-24
> >
> > https://github.com/bennyguo/instant-nsr-pl

---

> ### Author Response · Authors · 2024-11-24
> **Response to Reviewer tJmk**
>
> While it serves as a good reference, it may not offer a practical solution in application. In addition, the provided link is **not even a paper**, but an engineering project.

---

### Official Review · Reviewer_6ons · 2024-11-05

**Soundness:** 4
**Presentation:** 3
**Contribution:** 3
**Rating:** 6
**Confidence:** 4

**Summary:**

The paper presents a new dataset of 2.5k real cars scanned using modern
iPhones, resulting in high-resolution images as well as sparse LiDAR point
clouds (~200 views per car). This setting provides additional training and
benchmarking opportunities in novel view synthesis (NVS).

Key tasks which can be researched and benchmarked with this dataset include 3D
reconstruction, relighting, and parsing of car parts.

The dataset is further motivated with experiments demonstrating that using its
3D cars to perform image data augmentation can lead to perception improvements
in unusual scenarios. The dataset is also shown to help with part segmentation,
3D generative modeling, and 3D neural rendering.

One of my main questions regarding the paper, as elaborated in the Questions
Section, is whether any car is captured in more than one lighting condition. I
think this is a major research gap, so the presence of this data is pivotal in
assessing the impact of this dataset.

**Strengths:**

- [S1] The dataset includes diverse car types, including sedans, sports cars,
  and small trucks, captured in high resolution, and annotated with part
  information (e.g., doors, hood, wheels). This has applications in benchmarking
  3D reconstruction and in developing simulators for autonomous driving.
- [S2] The experimental section analyses multiple tasks such as neural rendering
  and 3D generative modeling, as well as camera-based perception in corner-case
  scenarios. In the latter example, the authors demonstrate that synthesizing
  corner-case images (e.g., cars swerving off road) using their proposed 3D car
  assets can help improve real data perception on these scenarios.
- [S3] The dataset is already openly available online!

**Weaknesses:**

- [W1] The LiDAR used is the iPhone 14 model, which is sparser than typical
  automotive LiDARs. While denser LiDAR can of course be re-simulated from dense
  3D models, this will require additional engineering effort while also
  suffering from some domain gap. This should be clarified in the intro: "3D
  scanner" makes me think of automotive or survey-grade LiDAR, not smartphone.
- [W2] Some minor suggestions for additional references and discussions: Please
  consider mentioning DeepMANTA [0] as related work since, while old, it also
  proposed similar part-level annotations.
- This isn't a weakness but as a tip: It can be helpful to provide some usage
  examples and tutorials for the dataset next to its SDK. For example "how to
  run gaussian splatting on 3DRealCar", since this can help more people learn to
  be familiar with the dataset, which can increase its impact.
- Other small suggestions:
  - For the "Stable Zero123" reference please surround the author name in curly
    braces in the .bib file, so the citation shows up as "Stability AI, 2023"
    instead of just "AI, 2023".
  - For those curious (such as myself!) it would be helpful to add some
    low-level details on the data collection in the appendix. For example, I
    have no idea how to get raw LiDAR points from an iPhone, so a brief
    discussion would be interesting.
  - L325: "the car is well-lighting" -> "the car is well-lit"
  - L417: "Dreamcract3D" contains a typo
- References:
  - [0]: Chabot, Florian, et al. "Deep manta: A coarse-to-fine many-task network
    for joint 2d and 3d vehicle analysis from monocular image." Proceedings of
    the IEEE conference on computer vision and pattern recognition. 2017.

**Questions:**

- [Q1] Is the proposed dataset restricted to academia or is commercial use
  allowed?
- [Q2] I looked at the sample car reconstruction provided on the 3DRealCar
  dataset, and noticed that the COLMAP output mesh, =textured_output.obj=, is
  missing the upper half of the car. Is this expected?
- [Q3] Are any of the vehicles in the dataset captured in more than one
  environment? It is valuable to have the same vehicle captured in multiple
  environments, with varying lighting conditions, because it creates ground
  truth for relighting. Relighting is challenging and it is otherwise very
  difficult to get non-synthetic ground truth, so having some of the cars
  captured in 2+ lighting conditions would be valuable. However, I cannot tell
  whether this is the case in the proposed dataset.
- [Q3] Will the dataset also include pre-computed dense reconstructions like
  splats, or dense meshes extracted from a NeRF-like method? As mentioned above, it
  seems like the COLMAP mesh outputs can be incomplete.
- [Q4] On L346, "2D Car Parsing", why are car parsing maps given as input to
  segmentation? Or does "S" refer to just a binary mask of the vehicle? (Vehicle
  vs. background?)

---

> ### Author Response · Authors · 2024-11-20
> **Response to Reviewer 6ons**
>
> We sincerely thank Reviewer 6ons for the valuable comments. We appreciate that Reviewer 6ons recognizes the strengths of our paper, including the 3DRealCar's unique contribution as a simulator. We are glad to see that Reviewer 6ons acknowledges the value of our proposed dataset and provides a positive rating. We address the concerns raised by Reviewer 6ons below.
>
> **Q1**: This "3D scanner" should be clarified as a smartphone.
>
> **A1:** Thank you for pointing out this problem. We will clarify this statement by clearly stating we use smartphones, as a scanner, to scan vehicles. Specifically, we adopt ARKit [1] and its API to collect images, depth maps, point clouds and reconstruct the CAD files with other camera parameters. Our data-collecting software enables phones to be a more convenient device to collect data. We will also release the code of our data collection software for phones.
>
>
> [1] https://developer.apple.com/augmented-reality/arkit/
>
>
> **Q2:** Referencing DeepMANTA [0] as related work since it also provides similar part-level annotations.
>
> **A2:** Thank you for suggesting this reference. The similarity of DeepMANTA with our dataset is that it also provides similar part-level annotations, but the difference is their annotations are based on synthetic CAD models while our dataset is based on real-world cars annotated by humans. We have cited this paper in lines 185-187 in our related work section, considering it has similarities with ours.
>
>
>
>
>
> **Q3:** Tip not weakness: provide some usage examples and tutorials for the dataset next to its SDK.
>
> **A3:** Thank you for your constructive suggestion. We will release all codes of this paper and provide detailed examples and guidance for other researchers to easily follow our work.
>
>
>
>
> **Q4:** Other small suggestions and typos.
>
> **A4:** Thank you for pointing out these typos. We will fix these problems in the revised version.
>
> Q5: How to get raw LiDAR points from an iPhone?
>
> A5: We directly adopt ARKit [1] to scan the target object. The raw LiDAR points can be obtained by two ways. The first is the scanned .obj file that can be used to generate LIDAR points. The second way is to use Colmap with multi-view feature matching for the attainment of LIDAR points.
>
> **Q5:** Is the proposed dataset restricted to academia or is commercial use allowed?
>
> **A5:** Yes, the proposed dataset allows academic usage. We will release all data for academic usage. Moreover, our released code is also allowed for academic usage.
>
>
> **Q6:** =textured_output.obj= is missing the upper half of the car.
>
> **A6:** This .obj file is generated by the ARKit API  [1]  rather than 3DGS. This result is in low-quality and cannot be directly used. We will explore to make fully use of this meta-data, like transforming it into point clouds for the initialization of 3DGS. On the contrary, the reconstructed 3D car assets are high-quality as shown in the project page demos.
>
>
>
> **Q7:** Having some of the cars captured in 2+ lighting conditions would be valuable.
>
> **A7:** Thank you for your constructive suggestion. Our dataset contains the cars of the same car type and the same color with different environmental lighting that is captured in different time and environments. For example, our dataset has BMW 530Li cars with same white color and different lighting conditions such as standard, reflective, and dark. We provide its Google Drive here: https://drive.google.com/file/d/1BP9BQQ7j-jvCzOIPXIvVJijl91qdm_jB/view?usp=sharing
> Our dataset can also be used as a real-world car relighting benchmark to increase its value.
>
>
>
>
> **Q8:** Will the dataset also include mesh generated by like splats, or a NeRF-like method?
>
> **A8:** Yes. To build a comprehensive dataset, especially meshes, we will provide mesh results in our dataset and release them to the public so that other researchers will follow our work more easily. We present some samples of the reconstructed meshes and material decompositions here.
> https://drive.google.com/file/d/11xU5wWemcc1g5OjZgdit3dNKIZiEyZKH/view?usp=sharing
>
>
> **Q9:**  On L346, "2D Car Parsing", why are car parsing maps given as input to segmentation? Or does "S" refer to just a binary mask of the vehicle? (Vehicle vs. background?)
>
> **A9:** Thank you for noticing this problem. We should clarify the input is an image and the output is a 2D car parsing map. "S" means multi-label segmentation maps as shown in Figure 1 with 13 car components.

---

> > ### Comment · Reviewer_6ons · 2024-11-23
> >
> > Thank you for your clarifications! The details about the use of ARKit are very helpful, given how good these new phones (and their software) are, perhaps more research groups should use them!

---

> > > ### Author Response · Authors · 2024-11-24
> > > **Response to Reviewer 6ons**
> > >
> > > Dear Reviewer 6ons
> > >
> > > We appreciate your constructive suggestions to further improve our work. We are wondering whether you have other concerns about our paper. We will be more than willing to answer your question.
> > >
> > > Best Regards,
> > > 3DRealCar Authors

---

> > > > ### Comment · Reviewer_6ons · 2024-11-25
> > > >
> > > > At this point, I have decided to maintain my original score.

---

### Author Response · Authors · 2024-11-20
**Thanks for All Reviewers and AC**

We appreciate the reviewers' insightful feedback. It is encouraging that all the reviewers recognize and agree that this paper presents a valuable 3D real car dataset including

**Community Contribution**:
- This dataset contains diversified car types (Reviewer 6ons, tJmk)
- This dataset provides detailed part-level annotations (Reviewer 6ons)
- The extensive experiments demonstrate the effectiveness of this dataset on various tasks.  (Reviewer 6ons,  tJmk, fi6m)
- This paper is well-organized and clearly presented  (Reviewer  tJmk)
- The proposed dataset is high-quality with comprehensive meta-data (Reviewer fi6m)
- The proposed dataset contains different lighting conditions to make up for the shortcomings of existing datasets (Reviewer fi6m)

**Since we are continually extending the scale of our dataset, we have collected more than 3000 cars in our dataset.  We will select 2500 high-quality data to release to the public.
We confirm our dataset and all codes will be released to the public to facilitate the development of the community.**

---

### Meta-Review · Area_Chair_m4nn · 2024-12-22

**Metareview:**

The paper introduces a large-scale, real-world 3D car dataset containing 2500 vehicles, each captured with around 200 RGB images and sparse LiDAR point clouds, using modern iPhones. The goal of the dataset is to advance research in vehicle reconstruction and modeling (e.g., NVS, part parsing, relighting, etc). The major strengths of the paper are the scale and diversity of the data. Comparing to existing real world dataset, it features more vehicles, more brands (vehicle types), and fine-grained part annotations. The data has the potential to improve existing perception systems, especially in corner cases. That being said, while the dataset itself is potentially interesting, the reviewers are also concerned about the core contributions of the paper --- the data curation pipeline is very standard and nothing innovative has been proposed, except applying it to more vehicles. Furthermore, while the authors did provide preliminary results on using the proposed data for certain downstream tasks, the experimental setups are not that comprehensive, casting doubts on the necessity of the dataset. The overall review of the paper is on the fence. While the reviewers appreciate the new data, they also point out that there are multiple additional experiments that can be done to make the paper more convincing and better justify the need of the data. After extensive discussion, the ACs decided to reject the paper. The authors are encouraged to re-submit the paper to a future venue.

**Additional Comments On Reviewer Discussion:**

The reviewers asked multiple questions on the dataset. For example, how can the dataset be used to learn relighting? The authors provide an example of a sedan being captured under different lighting conditions, showcasing the potential. While it is not clear if every single vehicle in the dataset is captured this way, at least some of them are. The authors also attempted to address the concerns on low-quality results as well as the necessity of the dataset. However, the ACs agree with the reviewers that the experimental setup can be further improved to more rigorously justify the effectiveness of the data (eg, While CADSim/GeoSim only reconstructs one side of the vehicle, their papers did show that they can improve perception too. Therefore, is it necessary to have full vehicle reconstructed? Also, more sophisticated experiments on NVS would definitely be helpful.)

---

### Decision · Program_Chairs · 2025-01-22

Reject